# Electroporation Delivery of Cas9 sgRNA Ribonucleoprotein-Mediated Genome Editing in Sheep IVF Zygotes

**DOI:** 10.3390/ijms25179145

**Published:** 2024-08-23

**Authors:** Wenhui Pi, Guangyu Feng, Minghui Liu, Cunxi Nie, Cheng Chen, Jingjing Wang, Limin Wang, Pengcheng Wan, Changbin Liu, Yi Liu, Ping Zhou

**Affiliations:** 1College of Animal Science and Technology, Shihezi University, Shihezi 832003, China; 13779708567@163.com (G.F.); 13201327891@163.com (M.L.); niecunxi@shzu.edu.cn (C.N.); enze105@163.com (C.C.); 2State Key Laboratory of Sheep Genetic Improvement and Healthy Breeding, Xinjiang Academy of Agricultural and Reclamation Sciences, Shihezi 832000, China; m13201327891@163.com (J.W.); wanglm1980@126.com (L.W.); wanpc@hotmail.com (P.W.); xlchangbin@163.com (C.L.); 3State Key Laboratory of Biocatalysis and Enzyme Engineering, School of Life Sciences, Hubei University, Wuhan 430062, China; yiliu0825@hubu.edu.cn

**Keywords:** electroporation, gene editing, CRISPR/Cas9, sheep zygotes, mutation efficiency, homologous recombination, in vitro fertilization (IVF)

## Abstract

The utilization of electroporation for delivering CRISPR/Cas9 system components has enabled efficient gene editing in mammalian zygotes, facilitating the development of genome-edited animals. In this study, our research focused on targeting the *ACTG1* and *MSTN* genes in sheep, revealing a threshold phenomenon in electroporation with a voltage tolerance in sheep in vitro fertilization (IVF) zygotes. Various poring voltages near 40 V and pulse durations were examined for electroporating sheep zygotes. The study concluded that stronger electric fields required shorter pulse durations to achieve the optimal conditions for high gene mutation rates and reasonable blastocyst development. This investigation also assessed the quality of Cas9/sgRNA ribonucleoprotein complexes (Cas9 RNPs) and their influence on genome editing efficiency in sheep early embryos. It was highlighted that pre-complexation of Cas9 proteins with single-guide RNA (sgRNA) before electroporation was essential for achieving a high mutation rate. The use of suitable electroporation parameters for sheep IVF zygotes led to significantly high mutation rates and heterozygote ratios. By delivering Cas9 RNPs and single-stranded oligodeoxynucleotides (ssODNs) to zygotes through electroporation, targeting the *MSTN* (Myostatin) gene, a knock-in efficiency of 26% was achieved. The successful generation of *MSTN*-modified lambs was demonstrated by delivering Cas9 RNPs into IVF zygotes via electroporation.

## 1. Introduction

Sheep play a crucial role in agricultural and societal systems, acting as a fundamental component of livestock farming and rural economies [1]. The utilization of genome editing methods has introduced a novel avenue for enhancing sheep breeding practices [2]. CRISPR-based technologies, recognized for their precise editing capabilities and user-friendly design, have emerged as the preferred method for achieving various genomic goals [3]. The CRISPR/Cas9 system has been effectively utilized to enhance sheep breeds by targeting genes that impact economically significant characteristics [4,5,6], as well as for generating large animal models [7,8].

Microinjection and electroporation are both effective techniques for delivering Cas9 RNPs into mammalian zygotes. While microinjection is known for its precision and reproducibility, it is labor-intensive and has a limited success rate, and is not suitable for high-throughput applications due to its costliness and the need for extensive training. Electroporation, on the other hand, is emerging as a more efficient and scalable method for genome editing, offering advantages such as reduced labor intensity and improved embryo survival and development rates [9,10]. Various animal models, including rats [11], mice [12,13], cattle [14], and pigs [15], have been successfully genetically modified using CRISPR/Cas9 components delivered via zygote electroporation.

The utilization of Cas9 ribonucleoprotein (RNP) by zygotes’ electroporation for genome editing has emerged as a widely adopted method due to its straightforward implementation and high efficiency. This investigation was centered on assessing the effectiveness of gene editing in sheep in vitro fertilization (IVF) zygotes through the application of the electroporation technique, with a specific focus on targeting the *ACTG1* and *MSTN* genes for knock-out or knock-in modifications. The *ACTG1* gene, responsible for encoding gamma-actin (Actin Gamma 1), plays a critical role as a highly conserved protein essential for various cellular motility functions and cytoskeleton maintenance [16]. Conversely, the *MSTN* gene, which functions to impede muscle differentiation and growth, is a well-documented target for enhancing muscle development in livestock [17]. Altering the *MSTN* gene has the potential to induce a double muscle trait, rendering it an appealing candidate for genome editing to enhance the production of lean meat in livestock [18]. Previous studies have demonstrated the modification of the *MSTN* gene to generate mutant sheep with increased body weight and muscle mass using the microinjection CRISPR/Cas9 system [2,19,20]. In a recent study, sheep blastocysts were successfully obtained through zygote electroporation [21] The present research endeavors to produce *MSTN*-modified lambs utilizing sheep IVF zygotes via the electroporation method.

It is imperative to empirically establish precise electroporation parameters in order to ensure high zygote viability rates and successful genetic modification across various species [10]. The optimization of voltage and pulse duration during the electroporation process is crucial for the specific cells and zygotes of each species [22,23]. For instance, Mahdi et al. determined the optimal conditions, involving a 40 V poring pulse, for the delivery of Cas9 RNPs into sheep zygotes [21]. Defining thresholds for the reversible impacts of electric fields during electroporation is essential for achieving effective genome editing in mammalian zygotes. Our study involved testing various poring voltages in the vicinity of 40 V and pulse durations for the electroporation of sheep zygotes.

Many research laboratories generally utilize frozen–thawed spermatozoa for in vitro fertilization (IVF) procedures due to their convenience and the potential for preserving genetic material. Studies have demonstrated that shorter incubation periods for ovine oocytes and spermatozoa during IVF can significantly improve the yield and quality of embryos [24]. One of the main challenges in gene editing of livestock zygotes using techniques like microinjection or electroporation is the increased occurrence of genetic mosaicism, which can hinder the consistency of genetic modifications [10,25]. It has been found that the timing of electroporation in relation to the cell cycle stage of the zygote is crucial for reducing mosaicism, with electroporation before the S-phase being more beneficial [26]. Delivering gene editing components prior to insemination through electroporation or microinjection has been shown to reduce mosaicism rates, thereby enhancing the uniformity of genetic alterations [27,28]. Effective production of gene knock-out blastocysts in sheep has been achieved by electroporating zygotes 6 h after fertilization, which optimizes editing efficiency and minimizes off-target effects [21]. We are establishing a reliable IVF protocol for frozen–thawed sheep semen, which has been provided ample experimental materials for electroporation-mediated gene editing of sheep zygotes.

The co-expression of Cas9 protein and sgRNAs in *E. coli* enables the concurrent generation of self-assembling Cas9 RNPs, denoted as “sf-Cas9 RNPs.” This technique presents a potentially cost-effective method for the production of Cas9 RNPs [29]. Our study aimed to utilize sf-Cas9 RNPs to ascertain the optimal conditions for electroporating sheep zygotes by assessing the efficacy of cleavage-stage embryos and blastocysts. While we successfully determined optimized electroporation parameters for sheep zygotes using sf-Cas9 RNPs, the resulting editing efficiency was unsatisfying. The purification of sf-Cas9 RNPs from a crude lysate may introduce impurities, such as bacterial RNA, which could detrimentally impact the genome editing activity of Cas9. This could lead to inconsistencies in enzymatic performance across different batches [30]. The efficacy of Cas9 RNPs, particularly the abundance of sgRNA, significantly influences the editing efficiency [31]. The incorporation of sgRNA into the Cas9 protein is recognized as the limiting factor for DNA cleavage, crucial for the overall effectiveness of the CRISPR/Cas9 system [32,33]. It is postulated that only a fraction of the Cas9 proteins in sf-Cas9 RNPs effectively bind sgRNAs, potentially due to various factors influencing the formation of the Cas9-sgRNA complex. The editing efficiencies of three formulations—sf-Cas9 RNPs alone, sf-Cas9 RNPs preincubated with synthetic sgRNAs (sf-Cas9 RNPs+sgRNA), and fully in vitro-assembled Cas9 RNPs (Cas9 RNPs)—were compared. Our findings revealed that a preincubation strategy involving different types of Cas9 protein and synthetic targeting sgRNAs enhanced subsequent editing activity by ensuring a greater number of Cas9 proteins were associated with sgRNAs before electroporation.

Co-transfection of Cas9 RNPs and short single-stranded DNA molecules, known as ssODNs, induced precise gene editing. ssODNs are engineered to match the target sequence and facilitate the incorporation of specific mutations [30,34,35]. These molecules act as a guide for the cell’s repair mechanisms when addressing DNA damage induced by Cas9 RNPs, thereby enabling the accurate introduction of mutations at specific genomic sites in sheep zygotes. In our study, we administered Cas9 RNPs and ssODNs designed for the *MSTN* gene in sheep, resulting in the precise integration of ssODN donor templates.

We utilized a consistent in vitro fertilization (IVF) technique in sheep to enhance the electroporation of zygotes for the delivery of the Cas9 RNP system, resulting in the generation of mutant sheep individuals. Our findings suggest promising prospects for the extensive utilization of gene editing technology in sheep breeding practices.

## 2. Results

### 2.1. Effect of Different Electroporation Settings on Cleavage and Blastocyst Rates

At 36 h post-fertilization, the cleavage percentages for the control group and the three experimental electroporation settings were 66.73% ± 9.83%, 68.84% ± 14.58%, 70.67% ± 6.98%, and 73.23% ± 13.17%, respectively (Figure 1). The differences were not statistically significant (*p* > 0.05), indicating that the electroporation settings did not adversely affect the cleavage rate. Although the cleavage rates between the control group and the experimental groups (electroporation settings 1, 2, and 3) were not significantly different, the average cleavage rate in the electroporated groups was slightly higher than in the control group. This slight increase might be attributed to the parthenogenetic activation effect caused by the electroporation of metaphase II (MII) oocytes and zygotes.

The blastocyst percentages for the control group and the three electroporation settings were 27.15% ± 10.98%, 23.18% ± 6.90%, 8.48% ± 2.23%, and 21.42% ± 7.82%, respectively (Figure 1). The significant reduction in the blastocyst rate for the group electroporated at 41 V for 3.5 ms (*p* < 0.05) highlighted the importance of optimizing electroporation parameters. The results suggested that electroporation setting 2 might be too aggressive and detrimental to the development of the zygotes into blastocysts.

The blastocyst development rates for electroporation settings 1 and 3 were not significantly different from the control group (*p* > 0.05). Both of these settings were more suitable for maintaining normal blastocyst development. These results suggested that the optimal combination of poring pulse voltage and pulse length for IVF zygotes of sheep was a voltage of 40~42 V and a pulse length of 3.0~3.5 ms.

Electroporation of zygotes showed a threshold-type phenomenon. A specific electric field strength (Eth) was required to trigger plasma membrane permeabilization in sheep zygotes. We concluded that stronger electric fields might require shorter pulse durations to achieve the desired effect without causing damage.

### 2.2. Effect of Three Formulations of Cas9 RNPs on the Mutation of Electroporation on ACTG1 Gene in Sheep

In the electroporation parameter optimization experiment, gene-edited blastocysts were produced using electroporation. However, the mutation efficiency was only 30.28% ± 9.14%, which was unsatisfactory.

The low mutation efficiency prompted the hypothesis that not all sf-Cas9 RNPs had successfully bound to sgRNAs. Proper binding was essential for the CRISPR/Cas9 system to target and edit specific genomic sequences. It was suggested that during the purification process, some sgRNAs had not been well protected, potentially affecting their functionality and, consequently, the overall mutation efficiency.

Another hypothesis was that contamination with bacterial RNA could have compromised the Cas9 genome editing activity. Contaminants interfered with the CRISPR/Cas9 system and led to reduced efficiency.

The experiment indicated that direct transfection of sf-Cas9 RNPs precomplexed with sgRNA using setting 1 electroporation improved gene editing efficiency. This suggested that the saturation of sgRNA in Cas9 RNPs could significantly impact the success of the gene editing process.

Comparisons of blastocyst editing efficiencies were made among three groups. The group with self-assembled Cas9 RNPs alone achieved an efficiency of 30.28% ± 9.14% (Figure 2). The group with sf-Cas9 RNPs + sgRNA resulted in a significantly higher efficiency of 87.78% ± 10.72% (Figure 2). The last group, with completely in vitro-assembled Cas9 RNPs, had an efficiency of 82.14% ± 15.57% (Figure 2).

The results highlighted the importance of the quality of Cas9 RNPs in achieving high mutation rates. We identified that precomplexing the Cas9 protein with sgRNA before electroporation could significantly enhance the mutation efficiency of gene editing zygotes.

### 2.3. Effect of Electroporation on MSTN Mutation Rates

The delivery of Cas9 RNPs via a series of electroporation treatments increased mutation efficiency. In the case of the sheep *MSTN* gene, the completely in vitro assembled Cas9 RNPs were transfected into zygotes using electroporation settings 1 and 3. The results demonstrated that electroporation significantly increased the mutation rate of the genomic target in sheep blastocysts (*p* < 0.01) (Figure 3). A very high percentage of blastocysts exhibited mutations in both the setting 1 and setting 3 groups, with mutation rates of 84.55 ± 1.19% and 82.83 ± 0.87%, respectively (Figure 3). These findings indicated that both sets of electroporation parameters were suitable for in vitro fertilized sheep zygotes. We again obtained the result that a higher pulse voltage required a shorter pulse length to achieve effective electroporation of zygotes.

### 2.4. Cas9 RNP-Mediated ssODN Knock-In Efficiency in Sheep Zygotes

A combination of Cas9 RNPs and ssODNs can introduce precise genetic alterations into ovine embryos through electroporation. To evaluate the introduction of ssODNs into a targeted genomic locus of ovine embryos via electroporation and Cas9 RNPs, IVF ovine zygotes were prepared. Electroporation setting 1 was utilized to deliver a mixture consisting of Cas9 protein (200 ng/μL), sgRNA (300 ng/μL) targeting the *MSTN* locus, and ssODNs containing *EcoR* I and *EcoR* V sites flanked by homology sequences (420 ng/μL) (Figure 4). Individual ovine blastocysts were lysed to extract template DNA solutions, which were subsequently used for PCR amplification. Genotyping of ovine blastocysts was performed using TIDER analysis based on the sequencing results of the PCR products.

The homologous recombination efficiency of the ssODN template was determined to be 26%, indicating that approximately one-quarter of the blastocysts successfully incorporated the ssODN at the targeted locus. Among the blastocysts analyzed, we only produced one blastocyst which was found to be homozygous for the ssODN introduction, while the rest were heterozygous. Representative images show the result of TIDER analysis and Sanger sequencing of one sample (Figure 4). These results indicated that the small DNA fragment could be efficiently inserted into the target region of sheep zygotes via electroporation with Cas9 RNPs.

### 2.5. Identification of MSTN Mutation Animals

Transferrable-grade sheep blastocysts, derived from zygote electroporation targeting the *MSTN* gene, were selected for embryo transfer. Five of these blastocysts were transferred into a single recipient ewe, which was on day 6 following CIDR (Controlled Internal Drug Release) device removal for estrous synchronization. Two male lambs were born as a result of this procedure (Figure 5). TIDE analysis was conducted to provide information on the mutation frequencies in these two lambs. Further analysis is required to determine whether the individual mutation types are heterozygous or chimeric.

TIDE analysis was performed to evaluate the gene editing outcomes at the *MSTN* locus in the two lambs. Both Lamb No. 1 and Lamb No. 2 displayed exceptionally high mutation rates of 94.5% and 94.9%, respectively (Figure 5). These results indicated that the CRISPR/Cas9 system was highly effective in inducing the intended genetic modifications in the lambs.

The proportion of wild-type alleles at the *MSTN* locus was recorded as zero for both lambs (Figure 5), suggesting that all alleles at this locus carried the induced mutation, with no unmodified (wild-type) alleles detected in the sequencing.

The successful editing of the *MSTN* gene, a key regulator of muscle growth, may have significant implications for enhancing muscle mass and improving production traits in sheep. The complete absence of wild-type alleles in these lambs could pave the way for new breeding strategies and genetic enhancements in agricultural contexts. This outcome is of great significance to the field of genetic modification in sheep husbandry.

## 3. Discussion

The CRISPR/Cas9 system can be delivered in three forms: as a plasmid, mRNA, and RNPs. Electroporation is an efficient method for delivering CRISPR/Cas9 mRNA or RNPs into mammalian embryos and requires specific conditions tailored to different mammalian zygotes. Voltage and pulse duration are critical parameters that need optimization for successful gene editing. Mahdi et al. established ideal parameters for incorporating Cas9 RNPs into sheep zygotes [21]. Further testing with different poring voltages and pulse lengths around the 40 V mark aimed to find the optimal combination for high gene mutation rates and reasonable blastocyst development. Our study determined that Settings 1 and 3 of the electroporation parameters were suitable for IVF zygotes of sheep. A weaker poring pulse voltage might necessitate a longer pulse duration to achieve the desired effect for electroporation mammalian zygotes.

Bacterial sf-Cas9 RNPs have demonstrated high gene editing efficiency both in vitro and in vivo [29]. In an experiment aimed at optimizing electroporation parameters for gene editing in sheep zygotes, sf-Cas9 RNPs were premixed with synthesized sgRNA to replicate the editing effect of fully assembled in vitro Cas9 RNPs. The results underscored the necessity for strict quality control in the purification process of sf-Cas9 RNPs derived from *E. Coli*, ensuring that the Cas9 protein and sgRNA were fully bound, which was critical for the effectiveness of gene editing.

Co-delivering Cas9 RNPs and ssODNs via electroporation can improve the consistency of mutagenesis [30]. An experiment utilized the *MSTN* gene locus in sheep embryos as the target site for gene editing, employing a 99-base ssODN as the homologous repair template to facilitate precise DNA repair after the Cas9 RNPs created a double-strand break. Sheep IVF zygotes were co-transfected with Cas9 RNP and ssODN templates by electroporation. The introduction of ssODNs into the blastocysts resulted in a 26% rate of homologous recombination, with a homozygote rate of 2% among these blastocysts. Increasing the length of the ssODNs could further improve the efficiency of homologous recombination, as longer ssODNs may provide a more stable and accurate template for the repair process, potentially leading to higher rates of successful gene editing.

We have established a stable procedure for embryo production by sheep IVF of frozen–thawed spermatozoa. Frozen–thawed spermatozoa are convenient for sheep IVF procedures. Anzalone et al. (2021) suggested that a shorter incubation period between oocytes and spermatozoa led to improved embryo production and quality [24]. This finding influenced IVF protocols to enhance outcomes. Mahdi et al. reported that performing electroporation on sheep zygotes 6 h after fertilization was an efficient method for producing gene editing knock-out blastocysts, with this timing being crucial for the successful delivery of the CRISPR/Cas9 system and subsequent gene editing [21]. Our experiments, confirming this timing, added weight to the suggested electroporation timing for sheep zygotes in gene editing protocols.

To confirm the validity of gene editing by electroporation in sheep zygotes, we randomly selected five blastocysts and transferred them into a recipient ewe. The transfer resulted in the successful birth of two male lambs, indicating that the gene editing process was successful in at least some of the transferred blastocysts. Mutations in the target *MSTN* gene were confirmed in both lambs, demonstrating that the gene editing was precise and effective. This suggested that electroporation of zygotes could be a genome editing and breeding tool for sheep husbandry.

Zygote electroporation technology could reduce research costs and improve animal welfare by creating gene-edited animals more efficiently. The use of electroporation as a delivery method for Cas9 RNPs is highlighted as a key technique in achieving these goals. The experiments also emphasize the importance of optimizing electroporation conditions for different loci and types of Cas9 RNPs to maximize gene editing efficiency in domestic animals.

Recent advancements in genome editing have greatly streamlined the creation of genetically modified animals. A novel study has integrated adeno-associated viruses (AAVs) with electroporation to establish a robust system for delivering CRISPR-Cas reagents into animal embryos [36,37]. Utilizing the full cargo capacity of an AAV, which is 4.7 kilobases, Chen et al. (2019) has successfully delivered long repair templates for intricate mouse genome engineering [38]. This is complemented by the electroporation of the Cas9 protein and sgRNAs complex, known as the AAV-EP method. The findings indicate that this combination of viral vectors and electroporation is not only less toxic but also inflicts minimal mechanical damage on the embryos. The AAV-EP technique offers an efficient avenue for executing large-scale and intricate genetic modifications.

Furthermore, the application of electroporation to sheep zygotes has streamlined the preparation of genetically altered sheep. The AAV-EP method’s capability to introduce lengthy DNA fragments and Cas9 RNP into sheep zygotes paves the way for the targeted integration of substantial genetic segments and the realization of complex genetic modifications.

The culmination of these experiments led to the successful creation of *MSTN* gene-edited sheep through embryo transfer, utilizing the optimized electroporation IVF zygote technique.

## 4. Materials and Methods

### 4.1. Experimental Design

A series of experiments were conducted to optimize the delivery of Cas9 RNPs into IVF sheep zygotes for gene editing via electroporation. The aim of Experiment 1 was to target the sheep *ACTG1* locus using sf-Cas9 RNPs and to evaluate the efficiency of gene editing at three different electroporation settings. Electroporation was performed 6 h post-fertilization. The objective was to identify the most efficient electroporation parameters. However, the mutation efficiency in sheep IVF blastocysts using sf-Cas9 RNPs was found to be unsatisfactory.

Experiment 2 compared the editing efficiencies of three different types of Cas9 RNPs delivered into sheep zygotes via electroporation. Experiment 3 built upon the findings from the first two experiments, aiming to confirm the efficiency of electroporation at another sheep locus, the *MSTN* gene, using the two most efficient electroporation settings identified previously.

Experiment 4 focused on testing the introduction efficiency of ssODN into a targeted genomic locus of ovine embryos using electroporation setting 1 and Cas9 RNPs.

Finally, in Experiment 5, *MSTN* gene-edited blastocysts were transferred to recipient ewes in an attempt to create gene-edited lambs using electroporation sheep IVF zygote technique.

### 4.2. Reagents and Solutions

All reagents and chemicals used in this study were purchased from FUJIFILM Wako Pure Chemical Corporation (Tokyo, Japan) unless otherwise stated. The restriction endonucleases used were *EcoR* I (1611, TAKARA, Beijing, China) and *EcoR* V (1612, TAKARA, Beijing, China). The electroporation solution was Opti-MEM (31985062, Thermo Fisher Scientific, Waltham, MA, USA).

The medium used for in vitro maturation (IVM) of cumulus–oocyte complexes (COCs) was based on bicarbonate-buffered TCM-199 (11150059, Thermo Fisher Scientific, Waltham, MA, USA), containing 2 mM glutamine, 10% (*v*/*v*) fetal bovine serum (FBS) (10100147C, Gibco), 5 μg/mL follicle-stimulating hormone (FSH) (Ovagen, ICP, Auckland, New Zealand), 5 μg/mL luteinizing hormone (LH), and 1 μg/mL β-estradiol.

The medium used for IVF and in vitro culture (IVC) was based on synthetic oviductal fluid (SOF) [39]. The IVF medium consisted of SOF supplemented with 5% (*v*/*v*) estrous sheep serum, 1 mM sodium pyruvate, 2 μg/μL heparin, 16 μM isoproterenol, and 2 mg/mL bovine serum albumin (BSA). The IVC medium was SOF supplemented with 0.4 mM sodium pyruvate, 1 mM glutamine, 0.28 mM inositol, 10 μg/μL gentamycin, 8 mg/mL BSA, MEM Non-Essential Amino Acids Solution (100X, 11140035, Thermo Fisher Scientific, Waltham, MA, USA), and MEM Amino Acids Solution EAA (50X, 11130051, Thermo Fisher Scientific, Waltham, MA, USA).

### 4.3. Oocyte Recovery and In Vitro Maturation (IVM)

Obtaining large numbers of mature oocytes and embryos from sheep through in vitro maturation (IVM), IVF, and in vitro culture (IVC) techniques is critical to the sheep industry. Briefly, sheep ovaries were collected from local slaughterhouses and maintained at 30 °C in a 0.9% NaCl solution to preserve their viability. These ovaries were then transferred to the laboratory within 4 h to ensure the freshness of the tissue.

The ovaries were exposed to 25 mM HEPES-buffered TCM-199 (HEPES-199, PM150612, Procell, Wuhan, China) containing 2% fetal bovine serum (FBS), 2 U/mL heparin, and 50 μg/mL gentamycin to create a suitable environment for follicle slicing. Follicles measuring 2–6 mm in diameter on the ovarian surface were sliced with a surgical blade. Under a stereomicroscope, cumulus–oocyte complexes (COCs) were collected and transferred into the IVM medium.

COCs were cultured in four-well dishes with 0.6 mL of IVM medium per well. Approximately 80 to 100 oocytes were placed in each well to mature in a humidified atmosphere at 38.5 °C with 5% CO_2_ for 24 h. After IVM, the COCs were observed under the stereomicroscope. Only those COCs showing an expansion of cumulus cells, indicating maturation, were selected for IVF.

The COCs were quickly pipetted into H199 medium and washed three times in pre-equilibrated IVF medium to remove any residual substances. The prepared COCs were then placed into 50 μL drops of IVF medium, with approximately 8 to 10 oocytes per drop, and covered with mineral oil to maintain optimal culture conditions.

### 4.4. In Vitro Fertilization (IVF)

The swim-up method was employed to select motile spermatozoa based on their ability to swim into the IVF medium. Frozen sheep semen was provided by the State Key Laboratory of Sheep Genetic Improvement and Healthy Breeding at the Xinjiang Academy of Agricultural and Reclamation Sciences, Shihezi, Xinjiang, China. The semen was frozen in 0.25 mL straws. For thawing, the straws were immersed in a water bath at 37 °C for 60 s.

For the swim-up procedure, 200 μL of frozen–thawed semen was layered under 1.5 mL of pre-equilibrated IVF medium in 2 mL tubes. The tubes were incubated for 50 min in an atmosphere of 5% CO_2_ in air. Following incubation, the uppermost 1.4 mL of medium, enriched with motile spermatozoa, was collected and centrifuged at 1200 rpm for 6 min. The supernatant was discarded, and the sperm pellet was resuspended in IVF medium to a concentration of 5 × 10^^6^ spermatozoa/mL. This suspension was then added to each drop of medium and incubated in a humidified atmosphere at 38.5 °C with 5% CO_2_ for six hours to allow for IVF to occur.

After fertilization, COCs were stripped by pipette aspiration in SOF-HEPES media and washed three times. The presumptive zygotes were then transferred to pre-equilibrated IVC medium and held until they were ready for electroporation treatment.

### 4.5. In Vitro Culture (IVC)

Presumptive zygotes from normal IVF or those subjected to electroporation were cultured together in groups of 80. Each group of zygotes was cultured in 600 μL of IVC medium, covered with 400 μL of mineral oil. The embryos were incubated in a humidified atmosphere at a temperature of 38.5 °C with 5% CO_2_ for seven days. Embryo development was assessed at 36 h for the 2-cell stage and on the 7th day post-IVF for blastocyst stage development.

### 4.6. sgRNAs

For the sheep *ACTG1* gene, a specific sgRNA was successfully used in a previous study [40]. This sgRNA was employed in our experiments to optimize the electroporation parameters for sheep zygotes, targeting the stop codon (Figure 2). *MSTN*, a negative regulator of muscle development, is commonly mutated in animals exhibiting the double-muscle phenotype, which is characterized by a significant increase in muscle mass. Consequently, *MSTN* is a preferred target gene for molecular breeding in farm animals to enhance muscle growth and meat yield. Crispo et al. reported the successful production of healthy myostatin knockout (KO) lambs using the CRISPR/Cas9 system via microinjection [5]. We utilized the same sgRNA to investigate the efficiency of the CRISPR/Cas9 system for editing the *MSTN* gene in sheep through electroporation, with the goal of generating KO sheep that promote muscle development and overall body growth (Figure 3 and Figure 4). The sgRNAs for these experiments were chemically synthesized on solid-phase support (Genebiogist, Shanghai, China) and solubilized to a concentration of 100 μM using enzyme-free water. Aliquots of 2 μL were prepared and stored at −80 °C until required for use.

### 4.7. Cas9 RNP Preparation

Three distinct formulations of Cas9 ribonucleoproteins (RNPs) were utilized to investigate gene editing in sheep zygotes via electroporation. The formulations were named as follows: sf-Cas9 RNPs, sf-Cas9 RNPs with supplementary sgRNA (sf-Cas9 RNPs+sgRNA), and Cas9 RNPs prepared by direct mixing.

Recently, Qiao et al. (2019) achieved co-expression of Cas9 enzymes and the associated sgRNA in *E. coli* to prepare self-assembling Cas9 RNPs, termed sf-Cas9 RNPs [24]. This method was cost-effective, leading to the adoption of sf-Cas9 RNPs for optimizing specific electroporation parameters. The pCold CL7-Cas9 co-expression plasmid was provided by Liu Lab of State Key Laboratory of Biocatalysis and Enzyme Engineering, School of Life Sciences, Hubei University. Sangon Biotech (Shanghai, China) constructed a specific plasmid targeting the sheep *ACTG1* gene based on the pCold CL7-Cas9 co-expression plasmid. DetaiBio (Nanjing, China) co-expressed and purified the sf-Cas9 RNPs, which had a concentration of 0.6 μg/μL in PBS solution. During electroporation, the final concentration of sf-Cas9 RNPs was adjusted to 120 ng/μL.

The initial use of sf-Cas9 RNPs did not yield satisfactory editing efficiency (Figure 2A). The quality of sf-Cas9 RNPs was suspected to contribute to the low editing efficiency, despite in vitro experiments indicating enzymatic activity.

To address this efficiency issue, a hypothesis-driven experiment was designed. One approach to improve the activity of sf-Cas9 RNPs was to supplement them with chemically synthesized sgRNA. A preincubation strategy was proposed, involving sf-Cas9 RNPs and synthetic targeting sgRNAs at a 1:2 molar ratio, allowing them to complex at room temperature for 10 min. This formulation was named sf-Cas9 RNPs+sgRNA. For electroporation delivery of sf-Cas9 RNPs+sgRNA, the final concentration of Cas9 RNPs was set to 120 ng/μL.

The third formulation, simply called ‘Cas9 RNPs’, involved the direct mixing of chemically synthesized sgRNA (Genebiogist, Shanghai, China) with recombinant Cas9 protein (A36498, Thermo Fisher Scientific, Waltham, MA, USA). This mixture was prepared at a molar ratio of 1:2 between sgRNA and Cas9 protein and incubated at room temperature for 10 min to facilitate the assembly of the Cas9-sgRNA complex. For the electroporation procedure, the final concentration of the Cas9 protein in the Cas9 RNP formulation was standardized to 250 ng/μL, ensuring consistent and effective delivery of the gene-editing machinery into the target zygotes.

### 4.8. ssODN

The ssODN templates were designed as 99-nucleotide sequences for homology-directed repair and synthesized with ULTRAPAGE purification (Sangon Biotech, Shanghai, China). The DNA oligonucleotides included sequences flanking the target region along with specific nucleotide changes that converted the target sequence (GCATGCTTGTGG) to one containing an *EcoR* I site (GAATTC) and an *EcoR* V site (GATATC), as depicted in Figure 4A. These ssODN templates were added to the electroporation mixture at a final concentration of 420 ng/µL.

### 4.9. Electroporation Programs Optimization

Optimizing electroporation programs for sheep zygotes involved adjusting several parameters to efficiently edit genes using CRISPR/Cas9 reagents targeting the *ACTG1* gene. Sheep oocytes and zygotes were electroporated using one of four different settings (Table 1). IVF zygotes in the control group were not electroporated, whereas those in the experimental groups underwent electroporation treatment. The transfer pulse was a common parameter across all experimental groups, with the following specifications: voltage at 5 V, pulse length of 50 ms, pulse interval of 50 ms, 5 pulses, a decay rate of 40%, and alternating polarity (+/−). The poring pulse varied among the experimental groups, particularly in voltage and pulse length. Setting 1 included 4 poring pulses of 3.5 ms at 50 ms intervals, starting at 40 V with a 10% decay rate and opposite polarity. Settings 2 and 3 were similar to Setting 1, except for the voltage and pulse length, which were set to 41 V and 3.5 ms, and 42 V and 3.0 ms, respectively. Electroporation was conducted on metaphase II (MII) oocytes and zygotes 6 h post-fertilization. The sf-Cas9 RNPs used targeted the *ACTG1* gene.

### 4.10. Electroporation

The NEPA21 Super Electroporator, equipped with Nepa Electroporation Cuvettes with a 1 mm gap (EC-001, Nepagene, Ichikawa-City, Japan), was utilized for the electroporation process. Denuded MII oocytes/zygotes were washed three times with pre-warmed Opti-MEM (31985062, Thermo Fisher Scientific, Waltham, MA, USA), then transferred to a fresh drop of Opti-MEM. They were mixed with RNPs to achieve a final volume of 20 µL and loaded to the bottom of the cuvette. The electroporation was performed using one of three different settings detailed in Table 1. Following electroporation, the recovered MII oocytes/zygotes were transferred to pre-equilibrated IVC medium and incubated for 30 min. Subsequently, these MII oocytes/zygotes were cultured in new 4-well dishes, each containing 0.6 mL of IVC medium and covered with 400 µL of mineral oil, for a period of 7 days.

### 4.11. Mutation Detection of Sheep Blastocysts

On the 7th day, sheep blastocysts were washed using a 10 µL pipette, sequentially passed through two droplets of DPBS (Dulbecco’s Phosphate-Buffered Saline). Each individual embryo was then transferred into a 0.2 mL PCR tube containing 10 µL of DNA extraction buffer (QE09050, Epicentre) using a pipette set to 1 µL. These PCR tubes were incubated at 65 °C for 15 min and then at 98 °C for 8 min. The extracted DNA was mixed with 12.5 µL of Taq PCR Master Mix (2X, with Blue Dye) (B639295, Sangon Biotech, Shanghai, China) and 1.0 µL of previously designed and tested primers (as detailed in Table 2). The first round of PCR was performed to amplify the DNA fragments flanking the targeted site, with conditions set for initial denaturation at 95 °C for 5 min, followed by 20 cycles of 95 °C for 30 s, 60 °C for 30 s, and 72 °C for 60 s. A nested PCR reaction continued to generate a highly specific PCR product, using a 1:10 dilution of the first PCR product. The second round of PCR was conducted under conditions similar to the first round but with 35 cycles. The product of the second round of PCR was separated by 1% agarose gel electrophoresis and compared to a control amplicon. Once the correct band was identified, the PCR product was sent for Sanger sequencing (Sangon Biotech, Shanghai, China) to determine the exact sequence around the targeted site. The editing efficiency was quantified using the TIDE version 3.3.0 (Tracking of Indels by Decomposition) and TIDER version 1.0.2 (Tracking of Insertions, Deletions, and Recombination events) software.

### 4.12. Embryo Transfer and Pregnancy Diagnosis

A total of 5 blastocysts, on the sixth day after IVF, derived from electroporation zygotes treated with Cas9 RNP targeting the *MSTN* gene, were transferred into one recipient ewe that had previously undergone estrous synchronization. The embryo transfer surgery was conducted under strict aseptic conditions, aided by laparoscopy (Henke Sass Wolf, Tuttlingen, Germany). Pregnancy diagnosis was performed 60 days post-IVF using transrectal B-mode ultrasonography with a 5 MHz probe (Well-D, Shenzhen, China). The recipient ewe was housed under standard conditions with ad libitum access to food and water. Ultimately, the lambs were born and genotyped for the induced mutations.

### 4.13. Identification of MSTN Mutation Lambs

Samples from the skin of two lambs were taken seven days after birth to identify and characterize knockout (KO) founders. Total DNA was isolated from the skin biopsies of all animals. PCR was performed using the same protocol as that for blastocyst samples. Genotyping of the *MSTN* exon 1 was conducted by direct sequencing of PCR amplicons using the sh*MSTN*F1 and sh*MSTN*R1 primers described earlier. Sanger capillary sequencing and the TIDE web tool were utilized for data analysis to evaluate the efficiency of the mutations.

### 4.14. Statistical Analysis

The percentages of zygote division, blastocyst formation, mutation occurrence, and ssODN introduction were analyzed using one-way ANOVA. When a significant difference was detected at *p* < 0.05, a Tukey’s range test was then employed to compare the means.

## Figures and Tables

**Figure 1 ijms-25-09145-f001:**
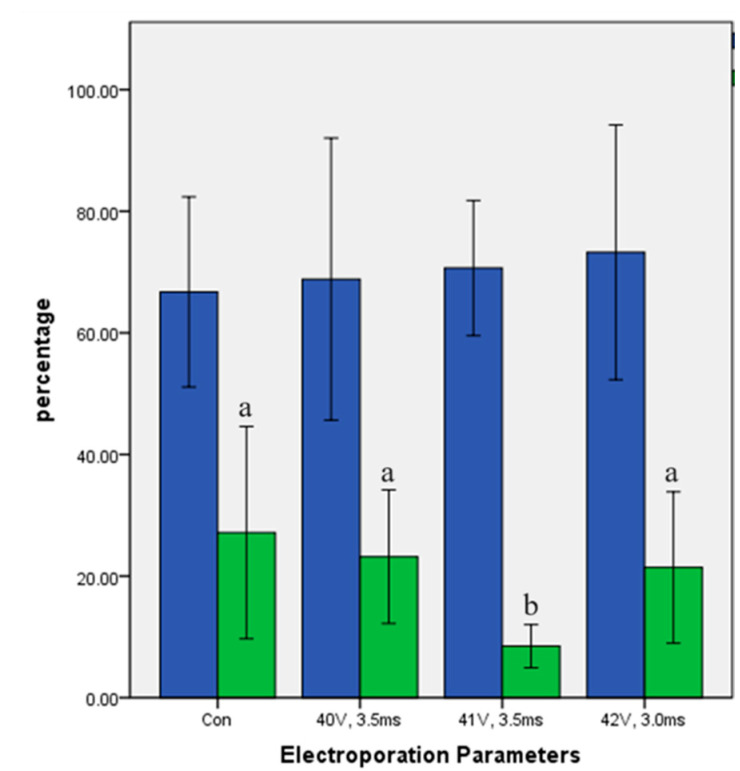
Cleavage and blastocyst rates following targeting to the *ACTG1* gene of sheep (means ± standard deviation). Means with different letters indicate significant differences (*p* < 0.05). Blue bars represent the cleavage rates post-IVF. Green bars represent the blastocyst formation rate.

**Figure 2 ijms-25-09145-f002:**
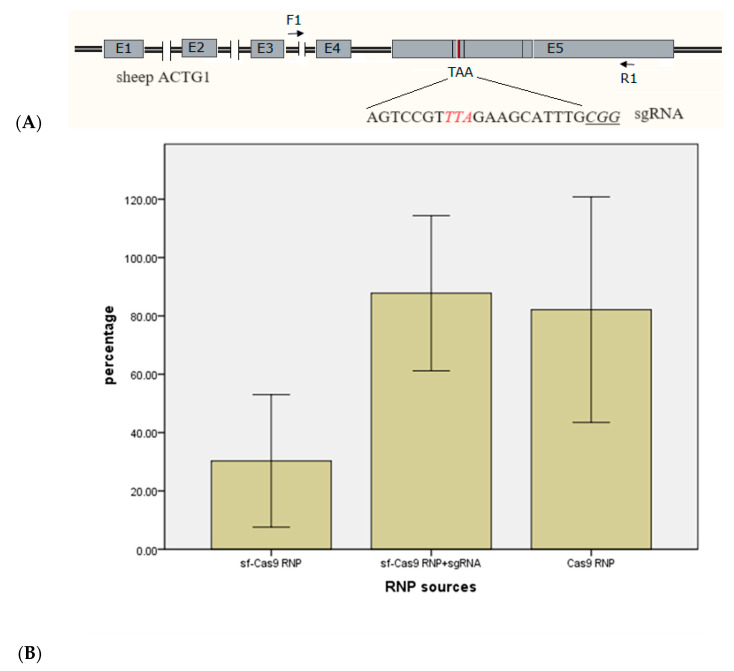
Effect of three formulations of Cas9 RNPs on blastocyst editing mutation rate at the *ACTG1* locus. Means with different letters indicate significant differences (*p* < 0.05). (**A**) The *ACTG1* gene in sheep and the sgRNAs used. Red letters were stop codons; underline was protospacer adjacent motif (PAM). (**B**) Mutation rate. Electroporation parameters were as follows: Poring pulse—voltage 40 V, pulse length 3.5 ms, pulse interval 50 ms, number of pulses 4, decay rate 10%, polarity +; Transfer pulse—voltage 5 V, pulse length 50 ms, pulse interval 50 ms, number of pulses 5, decay rate 40%, polarity +/−. Sf-Cas9 RNP: Co-expression and self-assembly of Cas9 protein and related sgRNAs in *E. coli*. Sf-Cas9 RNP + sgRNA: Preincubation of sf-Cas9 RNPs with synthetic targeting sgRNAs. Cas9 RNP: Completely in vitro-assembled Cas9 RNPs using commercial Cas9 protein and synthetic targeting sgRNAs.

**Figure 3 ijms-25-09145-f003:**
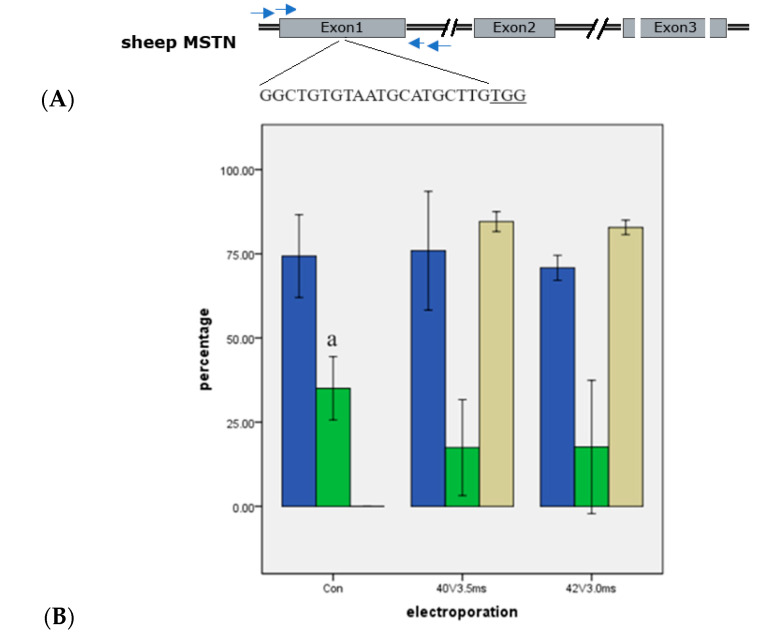
Mutation rate after targeting *MSTN* in sheep (means ± standard deviation). Means with different letters differ significantly (*p* < 0.01). (**A**) The targeting *MSTN* in sheep and sgRNAs. (**B**) Mutation rate. Con. was normally IVF group. 40 V 3.5 ms: Electroporation setting 1 (poring: voltage: 40 V, pulse length: 3.5 ms, pulse interval: 50 ms, pulse number: 4, decay rate 10% polarity: +. Transfer: voltage: 5 V, pulse length: 50 ms, pulse interval: 50 ms, pulse number: 5, decay rate 40%, polarity: +/−). 42 V 3.0 ms: Electroporation setting 3 (poring: voltage: 40 V, pulse length: 3.5 ms, pulse interval: 50 ms, pulse number: 4, decay rate 10% polarity: +. Transfer: voltage: 5 V, pulse length: 50 ms, pulse interval: 50 ms, pulse number: 5, decay rate 40%, polarity: +/−). Arrows were primers; underline was PAM.

**Figure 4 ijms-25-09145-f004:**
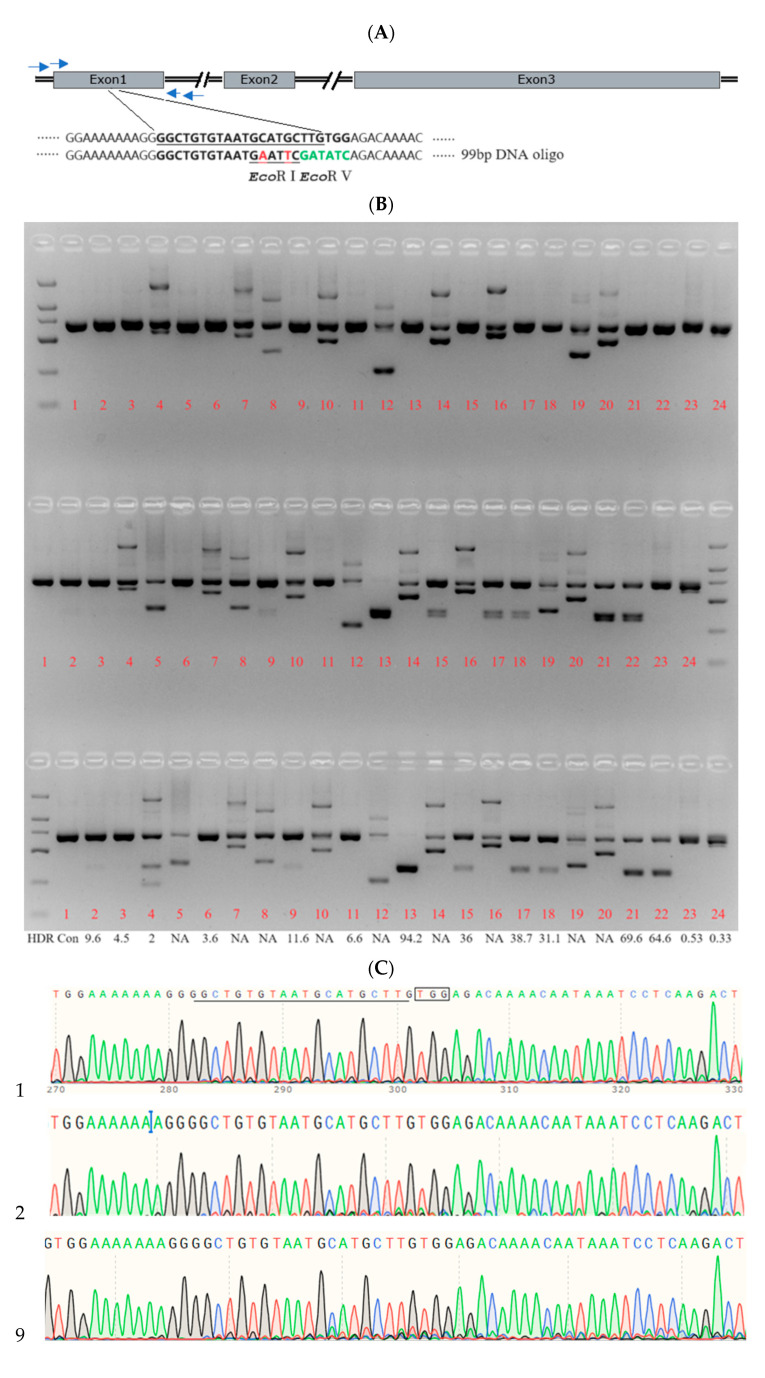
Correction of a homozygous mutation in the *MSTN* gene by electroporation of zygotes with Cas9 RNPs and ssODNs. (**A**) Schematic representation of the *MSTN* target sequence and the ssODN donor. The sgRNA target sequence is underlined. Nucleotide changes in the DNA oligo donor are highlighted in a different color. The *EcoR* I recognition site is indicated in red, and the *EcoR* V recognition site is shown in green within the DNA oligo donor. Arrows were primers. (**B**) RFLP (Restriction Fragment Length Polymorphism) analysis of 24 sheep blastocysts, comprising 1 blastocyst from the control group and 23 blastocysts from electroporation setting 1. HDR represents the knock-in efficiencies of the ssODNs by TIDER analysis. (**C**) Sequencing traces of PCR products spanning the *MSTN* target region for samples labeled in panel B, specifically B2, B9, B13, B15, B17, B18, B21, and B22. Clear sequencing traces suggest the predominance of a single allele (defined mutation) at the target locus, as exemplified in B13. In contrast, other lanes show a mix of wild-type and mutant alleles in the sheep blastocysts. ‘WT’ stands for wild type and ‘Con.’ denotes the control sample of a normal blastocyst.

**Figure 5 ijms-25-09145-f005:**
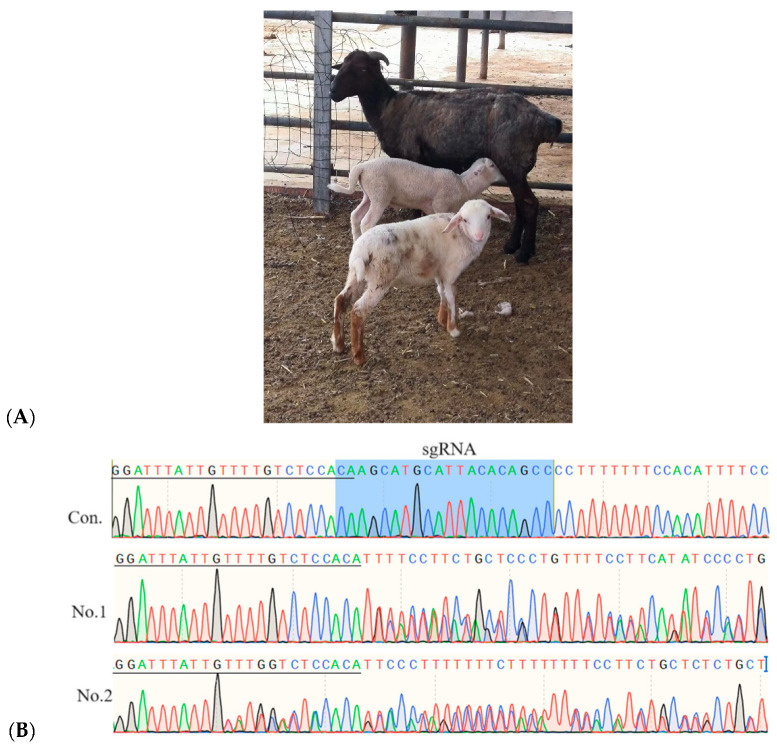
Photos of both lambs at 20 days after birth. Lamb No. 1 has a spotted coat, while Lamb No. 2 has a white coat. (**A**) A photograph of the recipient ewe with her two lambs. (**B**) Sequencing traces showing the *MSTN* targeting site for both lambs. Blue was the sgRNA sequence. The underline represented the consistent sequence from different sequencing results. (**C**) Quantitative analysis of indels around the sgRNA site for both lambs, utilizing the TIDE web tool.

**Table 1 ijms-25-09145-t001:** Electroporation parameters.

Scheme 1	Electroporation Parameters						
	Pulse Type	Voltages (V)	Pulse Length (ms)	Pulse Interval (ms)	Pulses Number	Decay Rate (%)	Polarity
Con.	——	——	——	——	——	——	——
1	Poring	40	3.5	50	4	10	+
	Transfer	5	50	50	5	40	+/−
2	Poring	41	3.5	50	4	10	+
	Transfer	5	50	50	5	40	+/−
3	Poring	42	3	50	4	10	+
	Transfer	5	50	50	5	40	+/−

**Table 2 ijms-25-09145-t002:** Primers for amplification of target fragments.

Primer	Scheme 5′–3′
sh*ACTG1*F1	AGCATGACTGACCTCCCTTTG
sh*ACTG1*R1	CCCAACCCCATGTAAGACCG
sh*ACTG1*F2	CACCATGTACCCTGGCAT
sh*ACTG1*R2	ACATTCTCACCTCAGCTAC
sh*MSTN*F1	GTGACTTGTGACAGACAGGGTT
sh*MSTN*R1	AATGTAGCAGCTTTCAGTCTCAT
sh*MSTN*F2	AATCACAGATCCCGACGACAC
sh*MSTN*R2	TCCTTACGTACAAGCCAGCAG

## Data Availability

Data are contained within the article.

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
