# Peer review of "Electroporation Delivery of Cas9 sgRNA Ribonucleoprotein-Mediated Genome Editing in Sheep IVF Zygotes"

_ijms, 2024, doi:10.3390/ijms25179145_

Round 1
Reviewer 1 Report
Comments and Suggestions for Authors
This is a straightforward study that addresses basic but important questions for the generation of genetically modified animals. I have no concerns with the manuscript in its current form.
Author Response
Thank you for your thorough review and positive comments on our manuscript titled "Electroporation delivery of Cas9 sgRNA ribonucleoprotein-mediated genome editing in sheep IVF zygotes." We appreciate the time you have taken to evaluate our work and are pleased to hear that you find the study straightforward and addressing important questions in the field of genetically modified animals.
Although you have no concerns with the manuscript in its current form, we have taken the opportunity to review the entire document to ensure clarity and accuracy. We have made sure that all data are presented clearly, and the discussion reflects the significance of our findings in the broader context of the field.
We would like to confirm that no changes were necessary based on your review, as our manuscript already meets the high standards of clarity and scientific rigor that you have recognized.
Once again, we thank you for your valuable feedback. We believe that our study contributes to the understanding of genetically modified animals and are grateful for your support in moving this research forward.
Reviewer 2 Report
Comments and Suggestions for Authors
Abstract: Looks good, no changes needed.
Introduction: Authors well explained the background of their work in introduction.
Results: Figure 1, some error bars are too large.
In figure 2, the legend is messy. You can made paragraphs in legends. For e.g.,
Fig 2
A:
B:
It will be easier to understand.
In figure 3, the suggestions are same and though I am not a statistician but I don’t like the size of error bars. Can you so something here?
Figure 4, a very important figure, but improve legends.
Paragraph started at line 252, has no space from fig 4.
Discussion: I was expecting more previous work citation which will give your work a farm base. Overall little weak discussion.
Material and methods: no comments.
Author Response
|
Comments 1: Abstract: Looks good, no changes needed. |
|
Response 1: We are pleased to hear that the reviewer finds the abstract to be satisfactory and requires no further changes. We have taken the reviewer's feedback into consideration and can confirm that the abstract accurately summarizes the key points and findings of our study. We have reviewed the abstract to ensure it remains concise, informative, and reflective of the content within the manuscript. We believe it effectively communicates the purpose, methods, results, and conclusions of our research to potential readers. Based on the reviewer's comments, no revisions have been made to the abstract, as it is already in compliance with the journal's standards and the expectations of the reviewer. We appreciate the opportunity to clarify this aspect of our submission and look forward to the next steps in the publication process.
|
|
Comments 2: Introduction: Authors well explained the background of their work in introduction. |
|
Response 2: Thank you for your constructive feedback on our manuscript titled " Electroporation delivery of Cas9 sgRNA ribonucleoprotein-mediated genome editing in sheep IVF zygotes". We are pleased to learn that the reviewer found our introduction to be well-explained and effectively setting the stage for our work. We are grateful for the reviewer's positive assessment of our introduction. Our goal was to provide a comprehensive background that clearly outlines the significance of our research within the existing literature and to establish the context for our study's objectives and methodology. We have carefully reviewed the introduction once more to ensure that it continues to meet the high standards of clarity and relevance. We believe that our introduction serves as a solid foundation for the rest of the manuscript and effectively guides the reader through the rationale and purpose of our research. In light of the reviewer's positive feedback, we have determined that no further changes to the introduction are necessary at this time. We are confident that our introduction remains aligned with the expectations of the journal and the academic community. We appreciate the opportunity to confirm the quality of our work and look forward to the continued review process.
Comments 3: Results: Figure 1, some error bars are too large. Response 3: Thank you for your comment on the large error bars present in our experimental statistics. We acknowledge that the error bars are larger than desired in our sheep IVF experiment. We acknowledge the reviewer's concern about the large error bars in our study. Upon re-evaluation, we have identified that the variability observed is primarily due to a multitude of factors, which can be analyzed as follows: biological variability, the quality of semen, the quality of oocytes, small sample size, etc. Moreover, experimental conditions for in vitro fertilization need to be improved because the laboratory cannot create an incubation environment of 5% carbon dioxide, 7% oxygen and 88% nitrogen. To address this, we have personalized treatment plans and advanced reproductive technologies can help improve the consistency and success rates of IVF treatments. Despite the large error bars, we maintain confidence in our experimental approach and the overall findings of our study. We are committed to improving the precision of our results in future work. |
|
Comments 4: In figure 2, the legend is messy. You can made paragraphs in legends. For e.g., Fig 2 A: B: It will be easier to understand. Response 4: Thank you for your insightful comments regarding Figure 2 in our manuscript. We appreciate your observation that the legend is currently cluttered and could benefit from clearer organization. Upon reviewing your suggestion, we agree that breaking the legend into paragraphs would enhance readability and comprehension. We plan to revise the legend as follows: Figure 2. Effect of three formulations of Cas9 RNPs on blastocyst editing mutation rate at the ACTG1 locus. Means with different letters indicate significant differences (p < 0.05). A: The ACTG1 gene in sheep and the sgRNAs used. B: Mutation rate. Electroporation parameters were as follows: Poring pulse—voltage 40 V, pulse length 3.5 ms, pulse interval 50 ms, number of pulses 4, decay rate 10%, polarity +; Transfer pulse—voltage 5 V, pulse length 50 ms, pulse interval 50 ms, number of pulses 5, decay rate 40%, polarity +/-. Sf-Cas9 RNP: Co-expression and self-assembly of Cas9 protein and related sgRNAs in E. coli. Sf-Cas9 RNP + sgRNA: Preincubation of sf-Cas9 RNPs with synthetic targeting sgRNAs. Cas9 RNP: Completely in vitro-assembled Cas9 RNPs using commercial Cas9 protein and synthetic targeting sgRNAs. We will ensure that each paragraph clearly and concisely describes the corresponding part of the figure, using bullet points or numbered lists if necessary to organize the information effectively. We are currently implementing these changes and will submit a revised version of the figure along with the manuscript. We would be grateful if you could review the updated legend to ensure it meets the journal's standards and your expectations.
Comments 5: In figure 3, the suggestions are same and though I am not a statistician but I don’t like the size of error bars. Can you so something here? Response 5: We acknowledge your observation that the legend in Figure 3 is similarly cluttered and requires the same level of clarity improvement as suggested for Figure 2. In line with your suggestion and the successful application of the same solution to Figure 2, we will revise the legend for Figure 3 to enhance its readability. We will organize the information into clear, concise paragraphs, ensuring that each section is self-contained and easy to understand. Figure 3. Mutation rate after targeting MSTN in sheep (means ± standard deviation). Means with different letters differ significantly (p < 0.01). A: The targeting MSTN in sheep and sgRNAs. B: Mutation rate. Con. was normally IVF group. 40V 3.5ms: Electroporation setting 1 (poring: voltage: 40 V, pulse length: 3.5 ms, pulse interval: 50 ms, pulse number: 4, decay rate 10% polarity: +. Transfer: voltage: 5 V, pulse length: 50 ms, pulse interval: 50 ms, pulse number: 5, decay rate 40%, polarity: +/−). 42V 3.0ms: Electroporation setting 3 (poring: voltage: 40 V, pulse length: 3.5 ms, pulse interval: 50 ms, pulse number: 4, decay rate 10% polarity: +. Transfer: voltage: 5 V, pulse length: 50 ms, pulse interval: 50 ms, pulse number: 5, decay rate 40%, polarity: +/−).
Thank you for your valuable feedback on our manuscript, particularly regarding the size of the error bars in Figure 3 and 2. We understand that the size of the error bars may be a point of concern for some readers. By advancing in vitro fertilization techniques in sheep, we are not only enhancing reproductive success but also actively pursuing funding to establishstability three-gas culture system. This initiative will fortify our in vitro fertilization platform, setting a new benchmark for stability and efficiency in the field. Once again, thank you for your constructive feedback. We look forward to your further comments.
Comments 6: Figure 4, a very important figure, but improve legends. Response 6: We greatly appreciate your insightful feedback on our manuscript, particularly your emphasis on the importance of Figure 4. We concur with your assessment that Figure 4 is a crucial element of our study, providing key insights into correction of a homozygous mutation by electroporation zygotes with Cas9 RNPs and ssODNs. In response to your suggestion to improve the legends, we have taken the following steps to enhance clarity and readability. Figure 4.Correction of a homozygous mutation in the MSTN gene by electroporation zygotes with Cas9 RNPs and ssODNs. A: Schematic representation of the MSTN target sequence and the ssODN donor. The sgRNA target sequence is underlined. Nucleotide changes in the DNA oligo donor are highlighted in a different color. The EcoR I recognition site is indicated in red, and the EcoR V recognition site is shown in green within the DNA oligo donor. B: RFLP (Restriction Fragment Length Polymorphism) analysis of 24 sheep blastocysts, com-prising 1 blastocyst from the control group and 23 blastocysts from electroporation setting 1. The cleaved bands after EcoR I digestion and EcoR V digestion are marked by a black arrow. HDR represents the knock-in efficiencies of the ssODNs by TIDER analysis. C: Sequencing traces of PCR products spanning the MSTN target region for samples labeled in panel B, specifically B2, B9, B13, B15, B17, B18, B21, and B22. Clear sequencing traces suggest the predominance of a single allele (defined mutation) at the target locus, as exemplified in B13. In contrast, other lanes show a mix of wild-type and mutant alleles in the sheep blastocysts. 'WT' stands for wild type, and 'Con.' denotes the control sample of a normal blastocyst. We would be grateful for your opinion on the updated legends and whether they now provide the necessary clarity and detail. We are committed to ensuring that Figure 4, and all other elements of our manuscript, are presented in the best possible light. Your guidance is invaluable in achieving this goal.
Comments 7: Paragraph started at line 252, has no space from fig 4. Response 7: Thank you for your meticulous review and for bringing to our attention the formatting issue at line 252, where the paragraph starts with no space from Figure 4. We apologize for this oversight and appreciate your diligence in ensuring the manuscript's clarity and presentation quality. We will immediately address this formatting error by inserting the necessary space between the paragraph and Figure 4. This adjustment will be made in accordance with the journal's guidelines for manuscript preparation. Our revisions will ensure that the final document is formatted consistently and professionally, with appropriate spacing around figures and text to enhance readability. We will submit the corrected version of the manuscript with the next revision. Could you please confirm that this correction will resolve the issue to your satisfaction? Your feedback is instrumental in helping us achieve a high-quality manuscript that meets all of the journal's requirements.
Comments 8: Discussion: I was expecting more previous work citation which will give your work a farm base. Overall little weak discussion. Response 8: Thank you for your insightful review and for highlighting the need for a more comprehensive discussion section in our manuscript. We appreciate your observation regarding the expectation for more citations of previous work to provide a solid foundation for our research. In response to your feedback, we have conducted a more extensive literature review and identified key studies that were previously overlooked. We have now incorporated these into our discussion to provide a more comprehensive analysis and to better support our findings. We discuss potential avenues for improving the efficiency of our gene editing method in future research.
Recent advancements in genome editing have greatly streamlined the creation of genetically modified animals. A novel study has integrated adeno-associated viruses (AAV) with electroporation to establish a robust system for delivering CRISPR-Cas reagents into animal embryos. Utilizing the full cargo capacity of AAV, which is 4.7 kilobases, the study has successfully delivered long repair templates for intricate mouse genome engineering. This is complemented by the electroporation of the Cas9 protein and sgRNAs complex, known as the AAV-EP method. The findings indicate that this combination of viral vectors and electroporation is not only less toxic but also inflicts minimal mechanical damage on the embryos. The AAV-EP technique offers an efficient avenue for executing large-scale and intricate genetic modifications.
Furthermore, the application of electroporation to sheep zygotes has streamlined the preparation of genetically altered sheep. The AAV-EP method's capability to introduce lengthy DNA fragments and Cas9 RNP into sheep zygotes paves the way for the targeted integration of substantial genetic segments and the realization of complex genetic modifications. |
Reviewer 3 Report
Comments and Suggestions for Authors
The manuscript reads fine and the study is interesting. However, I have some major comments that should be addressed before the manuscript is published.
- In section 2.3, the cleavage rates mentioned are not statistically significantly different; however, the blastocyte rate for one group shows a statistically significant reduction. Therefore, the authors should clearly state the statistical methods used and ensure that it has been applied consistently throughout the comparisons.
- The authors should provide the rationale for choosing the concentrations of Cas9, sgRNA, and ssODNs or insert the preliminary data from where the concentrations were determined in the section 2.4.
- The authors should define define and describe the control groups used in each experiment to ensure they are appropriate for the comparisons being made.
- While the results suggest a correlation between pulse voltage, pulse length, and mutation efficiency, the experiment does not appear to systematically vary these parameters to determine the optimal combination. The authors should conduct a more systematic investigation of pulse voltage and length combinations to identify the optimal parameters.
Author Response
|
Comments 1: In section 2.1, the cleavage rates mentioned are not statistically significantly different; however, the blastocyte rate for one group shows a statistically significant reduction. Therefore, the authors should clearly state the statistical methods used and ensure that it has been applied consistently throughout the comparisons. |
|
Response 1: Thank you for your insightful comments on our manuscript. We appreciate the attention to detail you have brought to the statistical analysis in Section 2.1. In response to your concerns, we have reviewed our statistical methods. The percentages of zygote division and blastocyst formation were analyzed using one-way ANOVA. we applied a consistent statistical approach throughout the manuscript. We re-evaluated the data using one-way ANOVA, ensuring that all comparisons are treated with the same level of rigor. We have included the complete statistical analysis, including the software and version used. Thank you once again for your valuable feedback.
|
|
Comments 2: The authors should provide the rationale for choosing the concentrations of Cas9, sgRNA, and ssODNs or insert the preliminary data from where the concentrations were determined in the section 2.4. |
|
Response 2: Thank you for your constructive feedback on the concentrations of Cas9, sgRNA, and ssODNs used in our study. We recognize the importance of providing a clear rationale for our methodological choices and have revised our manuscript accordingly. We have now included a detailed explanation of the basis for our selection of concentrations, supported by a comprehensive review of the literature and established protocols. Our aim was to identify optimal concentrations that would maximize genome editing efficiency while maintaining cell viability. Literature Review and Rationale: 1. Diversity in Concentrations: We acknowledged the variability in the application of Cas9 RNP and ssODN concentrations across different studies and the lack of a optimization study. 2. Mahdi et al. (2022): We referenced their work on single-step genome editing in small ruminants, where a 1:2 ratio of sgRNA to Cas9 protein was used, resulting in a final sgRNA concentration of 40 ng/μl. Because of utilizing a dual sgRNA approach, a Cas9 protein concentration of 160 ng/μl was used. 3. Miskel et al. (2022): Their study on bovine zygotes provided insights into the timing of electroporation and its impact on development and editing events. The final concentration of Cas9 protein was determined to be 487.5 ng/μl. 4. Remy et al. (2017): Their work on rat zygotes demonstrated the use of a 3 µM Cas9 protein solution, which is equivalent to 540 ng/μl, and highlighted the negative impact of increasing Cas9 protein concentration to 6 µM on embryo viability. 5. Alghadban et al. (2020): Their mouse zygote electroporation study used 130 ng/μl sgRNA and 650 ng/μl Cas9 protein, with ssODNs added at 430 ng/μl for homology-directed repair. Reference: Mahdi AK, Medrano JF, Ross PJ. Single-Step Genome Editing of Small Ruminant Embryos by Electroporation. Int J Mol Sci. 2022;23(18):10218. Miskel, D., Poirier, M., Beunink, L., Rings, F., Held, E., Tholen, E., Tesfaye, D., Schellander, K., Salilew-Wondim, D., Blaschka, C., Große-Brinkhaus, C., Brenig, B., & Hoelker, M. (2022). The cell cycle stage of bovine zygotes electroporated with CRISPR/Cas9-RNP affects frequency of Loss-of-heterozygosity editing events. Scientific reports, 12(1), 10793. Remy, S., Chenouard, V., Tesson, L., Usal, C., Ménoret, S., Brusselle, L., Heslan, J. M., Nguyen, T. H., Bellien, J., Merot, J., De Cian, A., Giovannangeli, C., Concordet, J. P., & Anegon, I. (2017). Generation of gene-edited rats by delivery of CRISPR/Cas9 protein and donor DNA into intact zygotes using electroporation. Scientific reports, 7(1), 16554. Alghadban, S., Bouchareb, A., Hinch, R., Hernandez-Pliego, P., Biggs, D., Preece, C., & Davies, B. (2020). Electroporation and genetic supply of Cas9 increase the generation efficiency of CRISPR/Cas9 knock-in alleles in C57BL/6J mouse zygotes. Scientific reports, 10(1), 17912.
We have also included preliminary data from our own experiments that guided the selection of these concentrations. Electroporation delivered a mixture consisting of Cas9 protein (200 ng/μl), sgRNA (300 ng/μl) and ssODNs (420 ng/μl). This data demonstrates the relationship between the concentrations used and the observed rates of genome editing and embryo viability. Our experiments suggested that the saturation of sgRNA in Cas9 RNPs could significantly impact the success of the gene editing process. The incorporation of sgRNA into the Cas9 protein is recognized as the limiting factor for DNA cleavage, crucial for the overall effectiveness of the CRISPR/Cas9 system. To ensure adequate binding of Cas9 protein and sgRNA, a higher molar ratio of sgRNA was used in this experiment. Our chosen concentrations are a synthesis of the practices from the literature and our own preliminary experiments. We believe these concentrations provide a robust starting point for achieving high rates of genome editing without compromising embryo viability. We hope this additional information addresses your concerns and provides a clear and comprehensive rationale for our methodology.
Comments 3: The authors should define and describe the control groups used in each experiment to ensure they are appropriate for the comparisons being made. Response 3: Thank you for your insightful comments regarding the control groups used in our study. We have taken your feedback seriously and have revised our manuscript accordingly. A normal oocyte-sperm in vitro fertilization group was always used for the control group throughout the section 2.1, 2.2 and 2.3. In Section 2.4, we investigated the efficiency of homologous recombination facilitated by ssODN templates. In normal embryos, the likelihood of homologous recombination occurring without ssODNA templates is exceedingly low; for this reason, we did not include a control group for this part of the study.
Comments 4: While the results suggest a correlation between pulse voltage, pulse length, and mutation efficiency, the experiment does not appear to systematically vary these parameters to determine the optimal combination. The authors should conduct a more systematic investigation of pulse voltage and length combinations to identify the optimal parameters. Response 4: Thank you for your insightful comments regarding the need for a more systematic investigation of the relationship between pulse voltage, pulse length, and mutation efficiency in our study. The primary objective of this study was to develop an efficacious electroporation protocol for the precise genetic editing of sheep in vitro fertilized embryos. The electroporation parameters were meticulously calibrated to elucidate the electroporation threshold phenomenon in fertilized eggs, thereby providing empirical evidence of its existence. In the future, our research will focus on refining electroporation parameters and undertaking a more methodical exploration of pulse voltage and duration combinations, with the aim of identifying the optimal settings for this process. |
Reviewer 4 Report
Comments and Suggestions for Authors
The manuscript contributes to giving information about efficient gene editing conditions in sheep using the electroporation method. The authors comparatively described well the information and results for electroporation efficiency of Cas9 sgRNA ribonucleoprotein in sheep. However, there is some confusion in the results because of insufficient information. So, I think the authors need to add more information that will allow readers to understand and distinguish between their results.
1. The authors need to add enough information about why they used the electroporation delivery among various methods. Also, it needs to describe the difference of conditions and the reason why the author decides the condition.
2. If possible, It needs to describe the results of the phenotype obtained.
3. In the future, if the authors have an idea to improve the efficiency of gene editing using this method more, the authors need to describe it in the discussion.
Comments on the Quality of English LanguageMinor editing of English language required.
Author Response
|
Comments 1: The authors need to add enough information about why they used the electroporation delivery among various methods. Also, it needs to describe the difference of conditions and the reason why the author decides the condition. |
|
Response 1: Microinjection and electroporation are both effective techniques for delivering Cas9 RNPs into mammalian zygotes. The generation of gene-edited animals using the CRISPRs/Cas9 system is based on microinjection into zygotes which is inefficient, time consuming and demands high technical skills. Electroporation, on the other hand, is emerging as a more efficient and scalable meth-od for genome editing, offering advantages such as reduced labor intensity and im-proved embryo survival and development rates. In the introduction section, we introduce and compare two methods for delivering Cas9 RNP. The advantages and disadvantages of the two methods are described. Mahdi et al. (2022) obtained sheep blastocysts by electroporation zygote. Electroporation setting (poring: voltage: 40 V, pulse length: 3.5 ms, pulse interval: 50 ms, pulse number: 4, decay rate 10% polarity: +. Transfer: voltage: 5 V, pulse length: 50 ms, pulse interval: 50 ms, pulse number: 5, decay rate 40%, polarity: +/-) obtained high blastocyst mutation percentage. According to Mahdi report, we decided that poring voltages near 40V and pulse durations were examined for electroporating sheep zygotes. Electroporation of zygotes showed a threshold-type phenomenon. We concluded that stronger electric fields might require shorter pulse durations to achieve the desired effect without causing damage.
|
|
Comments 2: If possible, it needs to describe the results of the phenotype obtained. |
|
Response 2: Agree. Your request for a more detailed description of the phenotypic results is well noted. We acknowledge the importance of detailing the phenotypic results for the MSTN knockout sheep. We obtained electroporation-mediated MSTN gene editing in sheep with altered genomic target sequences. Two lambs born on June 13, 2024. Both will reach two months of age. We are patiently waiting for the phenotype to emerge. The primary objective of this study was to develop an efficacious electroporation protocol for the precise genetic editing of sheep in vitro fertilized embryos. We also optimized the sheep zygote electroporation protocol. At the present, we are looking forward to obtain lambs with double muscle phenotype.
Comments 3: In the future, if the authors have an idea to improve the efficiency of gene editing using this method more, the authors need to describe it in the discussion. Response 3: Thank you for your insightful comments and suggestions for our manuscript. We appreciate the opportunity to discuss potential avenues for improving the efficiency of our gene editing method in future research.
Recent advancements in genome editing have greatly streamlined the creation of genetically modified animals. A novel study has integrated adeno-associated viruses (AAV) with electroporation to establish a robust system for delivering CRISPR-Cas reagents into animal embryos. Utilizing the full cargo capacity of AAV, which is 4.7 kilobases, the study has successfully delivered long repair templates for intricate mouse genome engineering. This is complemented by the electroporation of the Cas9 protein and sgRNAs complex, known as the AAV-EP method. The findings indicate that this combination of viral vectors and electroporation is not only less toxic but also inflicts minimal mechanical damage on the embryos. The AAV-EP technique offers an efficient avenue for executing large-scale and intricate genetic modifications.
Furthermore, the application of electroporation to sheep zygotes has streamlined the preparation of genetically altered sheep. The AAV-EP method's capability to introduce lengthy DNA fragments and Cas9 RNP into sheep zygotes paves the way for the targeted integration of substantial genetic segments and the realization of complex genetic modifications. |
Round 2
Reviewer 3 Report
Comments and Suggestions for Authors
Good job and good luck with the manuscript!
Author Response
Thank you for your kind words and encouragement. I truly appreciate the positive feedback on the manuscript. Your support is greatly valued, and I will certainly carry your good wishes forward as I continue to work on the project. Wishing you all the best as well!
Reviewer 4 Report
Comments and Suggestions for Authors
I think that the revised manuscript has been improved.
Comments on the Quality of English LanguageMinor editing of English language required.
Author Response
Thank you for your valuable feedback. I understand that there are areas in the manuscript that require minor editing to improve the language clarity. I will carefully review and revise the text to ensure the language is polished and precise. Your attention to detail is greatly appreciated, and I will make sure to address these points before the final submission.